

# White-handed gibbons discriminate context-specific song compositions

Julie Andrieu[1], Samuel G. Penny[1,2], Hélène Bouchet[3],
Suchinda Malaivijitnond[4,5], Ulrich H. Reichard[6] and
Klaus Zuberbühler[1,3]

[1] Department of Comparative Cognition, University of Neuchâtel, Neuchâtel, NE, Switzerland
[2] School of Pharmacy and Biomolecular Sciences, University of Brighton, Brighton, UK
[3] School of Psychology and Neuroscience, University of St. Andrews, St. Andrews, UK
[4] National Primate Research Center of Thailand, Chulalongkorn University, Saraburi, Thailand
[5] Department of Biology, Faculty of Science, Chulalongkorn University, Bangkok, Thailand
[6] Department of Anthropology and Centre for Ecology, Southern Illinois University at Carbondale, Carbondale, IL, USA

## ABSTRACT

White-handed gibbons produce loud and acoustically complex songs when interacting with their neighbours or when encountering predators. In both contexts, songs are assembled from a small number of units although their composition differs in context-specific ways. Here, we investigated whether wild gibbons could infer the 'meaning' when hearing exemplars recorded in both contexts (i.e. 'duet songs' vs. 'predator songs'). We carried out a playback experiment by which we simulated the presence of a neighbouring group producing either its duet or a predator song in order to compare subjects' vocal and locomotor responses. When hearing a recording of a duet song, subjects reliably responded with their own duet song, which sometimes elicited further duet songs in adjacent groups. When hearing a recording of a predator song, however, subjects typically remained silent, apart from one of six groups which replied with its own predator song. Moreover, in two of six trials, playbacks of predator songs elicited predator song replies in non-adjacent groups. Finally, all groups showed strong anti-predator behaviour to predator songs but never to duet songs. We concluded that white-handed gibbons discriminated between the two song types and were able to infer meaning from them. We discuss the implications of these findings in light of the current debate on the evolutionary origins of syntax.

Corresponding author
Julie Andrieu,
julie.andrieu@unine.ch

## INTRODUCTION

Primate vocal communication is characterised by species-specific repertoires of acoustically distinct vocalisations, some of which are given in response to specific events. The classic example is the vervet monkey (*Chlorocebus pygerythrus*) alarm call system, with acoustically distinct call types given to different predator types (*Seyfarth, Cheney & Marler, 1980a*, *1980b*). However, beyond the fact that primate calls can convey relatively distinct meanings, additional complexities have recently come to light, with corresponding implications for evolutionary theories of communication.

First, it is often difficult to characterise a particular call type as an acoustically discrete structural entity. Instead, following in-depth investigation seemingly 'discrete' calls often display considerable amounts of acoustic variation, which may be meaningful to recipients (*Keenan, Lemasson & Zuberbühler, 2013*). For example, the acoustic structure of chimpanzee (*Pan troglodytes*) rough grunts varies depending on the perceived quality of the food resource (*Slocombe & Zuberbühler, 2005*), whereas Barbary macaque (*Macaca sylvanus*) barks differ in call duration and mean frequency range according to specific external disturbances (*Fischer, Hammerschmidt & Todt, 1995, 1998; Fischer & Hammerschmidt, 2006*).

Second, context can play an important role in how animals interpret each other's calls. Evidence is in terms of how ongoing context modifies how animals react to a specific call type (*Zuberbühler, 2000a, 2000b; Arnold & Zuberbühler, 2013; Seyfarth & Cheney, 2018*), a mechanism already described by *Smith (1977)*. Empirically, the way intention and external factors affect how primates infer meaning from signals is relatively poorly explored (*Grice, 1969; Carnap, 1988; Scott-Phillips, 2010*).

Third, call sequences can serve as powerful semantic vehicles beyond the contribution of individual calls (*Zuberbühler, 2019a*). For instance, the number of roaring units per sequence in guereza colobus monkey (*Colobus guereza*) alarm roars depends on the nature of the danger (*Schel, Tranquilli & Zuberbühler, 2009*). Another example is Campbell's monkeys (*Cercopithecus campbelli*) alarm calling, with variation in call rates (*Lemasson et al., 2010*), call combinations (*Ouattara et al., 2009*) and call permutations (*Ouattara, Lemasson & Zuberbühler, 2009a, 2009b*) depending on external events. Similar phenomena have been observed in putty-nosed monkeys (*Cercopithecus nictitans martini*) (*Arnold & Zuberbühler, 2006a, 2006b*). Although these findings show remarkable similarities to some aspects of human syntax in terms of combinatorial and permutational properties, the implications for evolutionary theories of language are far from clear, suggesting that more empirical work is needed (*Bolhuis et al., 2018; Townsend et al., 2018; Zuberbühler, 2019b*).

A relevant primate example of complex combinatorial structure is gibbon song. In most species, mated pairs produce morning duets that appear to serve territorial and mate defence functions (*Haimoff, 1984; Raemaekers & Raemaekers, 1985a; Geissmann, 2002; Terleph, Malaivijitnond & Reichard, 2015, 2016*; J. Andrieu, 2012–2014, unpublished data a). Social learning seems to play some role in the acquisition of song (*Koda et al., 2013*) and production is subject to social influence (e.g. changes in mating partners usually result in audible differences in song coordination) (*Geissmann, 1999; Terleph, Malaivijitnond & Reichard, 2017*). Like most other primate calls, gibbon song contains information about caller identity (*Oyakawa, Koda & Sugiura, 2007; Terleph, Malaivijitnond & Reichard, 2015; Clink et al., 2017*) and the caller's physical condition (*Barelli et al., 2013; Terleph, Malaivijitnond & Reichard, 2016*). Gibbon songs are audible over long distances, up to 1 km, much beyond an average gibbon home range (*Mitani, 1985*), suggesting that they have evolved to communicate to outgroup individuals (*Raemaekers & Raemaekers, 1985a, 1985b; Mitani, 1985; Terleph, Malaivijitnond & Reichard, 2015, 2016*).

Interestingly, in white-handed gibbons (*Hylobates lar*), there is also evidence for context-specific song types: duet songs are produced by the mated pair as part of their daily routine while predator songs are given when facing a predator, such as a clouded leopard or python (*Clarke, Reichard & Zuberbühler, 2006*; J. Andrieu, 2012–2014, unpublished data b). Both song types are identical in terms of their note repertoires, although there are consistent differences in the prevalence of certain notes and in how notes are combined into songs (*Clarke, Reichard & Zuberbühler, 2006*, *2015*). Predator songs are sung for longer than duet songs and produced by most group members. They function to deter predators, recruit group members, and alert relatives in adjacent territories (*Zuberbühler, Jenny & Bshary, 1999*; *Clarke, Reichard & Zuberbühler, 2006*; *Matsudaira et al., 2018*). In contrast, duet songs function in mate and territorial defence (*Marshall & Marshall, 1976*; *Raemaekers & Raemaekers, 1985a*). Duet songs may also function as indicators of the strength of the social bond of the mated pair, a kind of relationship marker, evidenced by the fact that newly formed pairs appear to go through a lengthy phase of adjusting their relative vocal contributions towards a well-adjusted duet song (*Haimoff, 1984*; *Geissmann & Orgeldinger, 2000*).

Here, we investigated whether gibbons could discriminate the two functionally and structurally distinct song types (i.e. duet song and predator song), by broadcasting natural singing events of a neighbouring group simulated from a concealed speaker. We predicted that if gibbons discriminated between predator and duet songs then they should respond with the matching song types and with behaviour adequate to the situation. Specifically, in response to predator songs we predicted increased vigilance, increased defaecation rates and any other type of anti-predator behaviour already reported in the literature (*Boissy, 1995*; *Clarke, Reichard & Zuberbühler, 2012*). In response to duet songs, we predicted no changes to antipredator behaviour but duet song responses (*Raemaekers & Raemaekers, 1985a*, *1985b*).

## MATERIALS AND METHODS

### Study site and subjects

This study was conducted in the Mo Singto-Klong E-Tau area of Khao Yai National Park, Thailand (101°22′E, 14°26′N), 130 km North-East of Bangkok. Data were collected from December 2012 to August 2014. Thirteen fully habituated groups of white-handed gibbons were monitored, each comprising a primary male, his mated female with her offspring and (in 5 cases) a secondary male, totalling $N = 53$ individuals at the time of the study. Due to a number of constraints, it was only possible to conduct playback experiments with six of the 13 groups (Table 1).

### Terminology

Following *Raemaekers, Raemaekers & Haimoff's (1984)* terminology we distinguished three sequence types within each song: the introductory sequence (series of soft 'hoo' notes, followed by combinations of other note types, such as 'oo', 'wa', 'leaning wa' and 'wa-oo'); the great call sequence (idiosyncratic female call sequence, usually followed by

**Table 1 Composition of study groups at the Mo Singto-Klong E-Tau research area (August 2014).**

| Group | N individuals | Group composition | Tested | Song provider for |
|---|---|---|---|---|
| A* | 3 | 2AM, 1AF | – | group H |
| B* | 5 | 2AM, 1AF, 1JF, 1I? | yes | – |
| BD | 3 | 1AM, 1AF, 1I? | – | – |
| C | 3 | 1AM, 1AF, 1I? | – | – |
| E | 3 | 1AM, 1AF, 1JM | – | – |
| H† | 4 | 1AM, 1AF, 1JF, 1I? | yes | – |
| M | 5 | 1AM, 1AF, 1SAF, 1JM, 1I? | yes | group R |
| N* | 6 | 2AM, 1AF, 1SAM, 1JF, 1I? | yes | group M |
| NOS* | 5 | 2AM, 1AF, 1J?, 1I? | – | – |
| R | 4 | 1AM, 1AF, 1AF, 1I? | yes | – |
| S | 3 | 1AM, 1AF, 1JM | – | group W |
| T | 5 | 1AM, 1AF, 1SAM, 1JM, 1I? | – | group B |
| W* | 4 | 2AM, 1AF, 1I? | yes | group N |

Notes:
* Multi-male group; M, male; F, female, ?, sex unknown; A, Adult (age > 8 years); SA, sub-adult (5–8 years); J, juvenile (2–5 years); I, infant (<2 years). yes, tested group; -, group not tested.
† No data on latency and duration of first look to speaker due to technical problems (duet playback: female filmed erroneously; predator playback: male moved out of sight).

her male's 'coda' response). In duet songs, the first great call sequence usually appears within the first 2 min. Great call sequences can be repeated multiple times (about once every 1–2 min) (*Raemaekers, Raemaekers & Haimoff, 1984*; *Clarke, Reichard & Zuberbühler, 2006*; *Terleph, Malaivijitnond & Reichard, 2016*), in which case they are separated by an interlude sequence (any notes given after a great call sequence, including the final one) (*Ellefson, 1968*; *Raemaekers, Raemaekers & Haimoff, 1984*) (Figs. 1 and 2A).

The same three sequence types can also be found in predator songs although, overall, they differ in length and are produced with the contribution of most group members. When comparing predator songs with duet songs, for the introductory sequence the initial 'hoo' notes series are longer and contain more 'hoo' notes, followed by fewer 'leaning wa' notes and more 'hoo' notes (*Clarke, Reichard & Zuberbühler, 2006*). The great call sequence is also different, mainly because males respond more rapidly with their answering coda (*Clarke, Reichard & Zuberbühler, 2006*). Regarding the interlude sequence, predator songs contain more 'sharp wow' notes, especially towards the end of the song, compared with duet songs (*Clarke, Reichard & Zuberbühler, 2006*) (Figs. 1 and 2B).

## Stimulus collection

Duet songs were recorded on an all-occurrence basis during all-day follows of study groups (Table 1) until at least one song suitable as playback stimulus was recorded, that is, a high-quality song with minimum background noise, singing individuals at a maximum distance of 30 m from the recording device. Predator songs were induced by presenting a realistic, life-size clouded leopard (*Neofelis nebulosa*) model to each group following an established protocol (Fig. 3, *Clarke, Reichard & Zuberbühler, 2006*). Once a group was

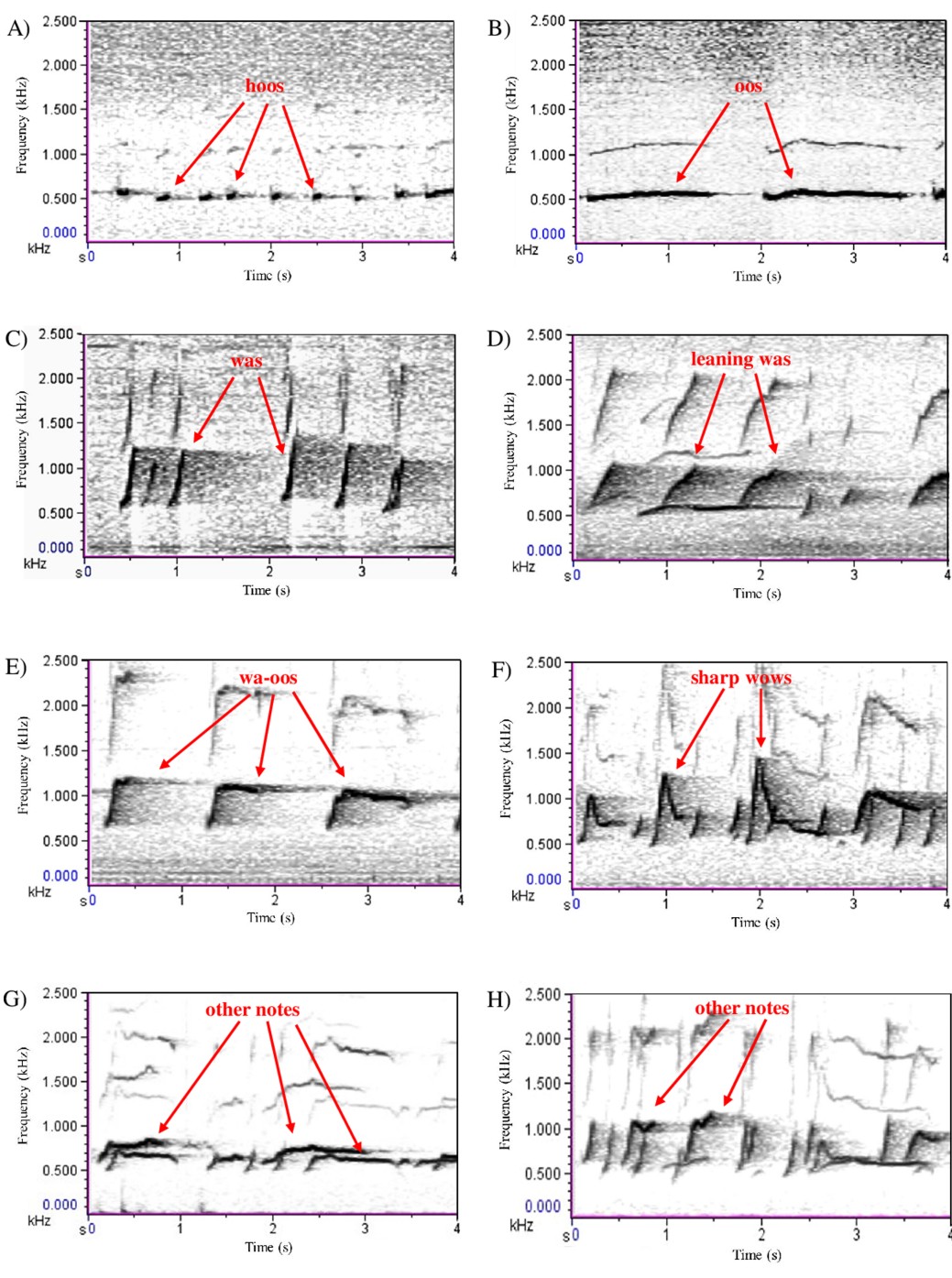

**Figure 1 Song note repertoire of white-handed gibbons (*Raemaekers, Raemaekers & Haimoff, 1984*; *Clarke, Reichard & Zuberbühler, 2006*).** Note types (A) 'hoo'; (B) 'oo'; (C) 'wa'; (D) 'leaning wa'; (E) 'wa-oo'; (F) 'sharp wow'; (G) 'other'; (H) 'other'. Songs were digitised using Cool Edit Pro 2.1; spectrograms were drawn using 21.6 Hz filter bandwidth, 2.69 Hz frequency resolution, 33.3 ms time grid resolution and a Hanning window function.

located and before positioning the model, we ensured that on the same day the group had (a) already produced at least one duet song more than one hour earlier (to verify a basic motivation to sing), (b) not yet produced a predator song (nor its direct neighbours),

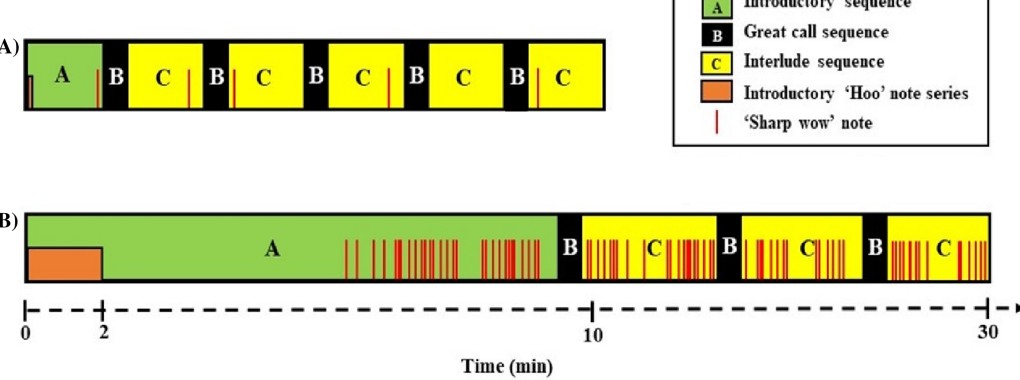

**Figure 2 Schematic representation of the structural differences between (A) duet and (B) predator songs (*Clarke, Reichard & Zuberbühler, 2006*).**

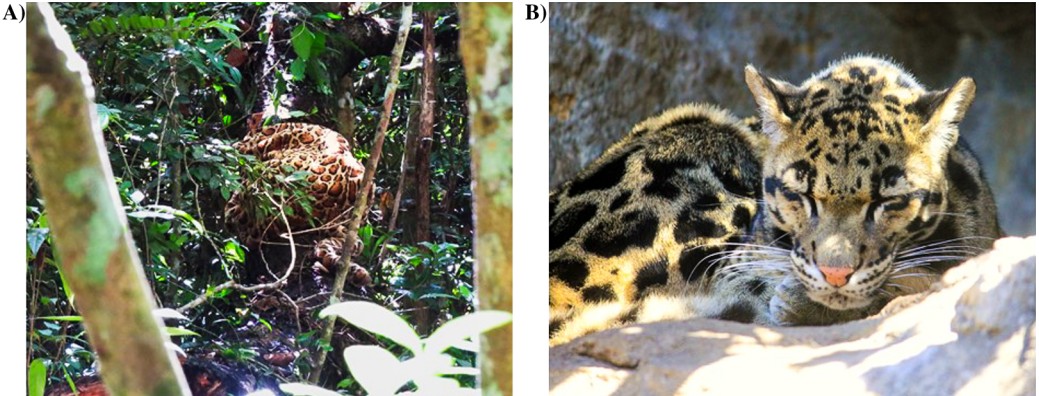

**Figure 3 (A) Clouded leopard model used to elicit predator songs (Photo credit: Julie Andrieu); (B) real clouded leopard, *Neofelis nebulosa* (Image credit: goodfreephotos.com at https://www. goodfreephotos.com/animals/mammals/clouded-leopard.jpg.php).**

(c) not had a natural predator encounter since the beginning of the day-follow, nor heard other species' alarm calls within the last hour and (d) not had an intergroup encounter with a neighbouring group. If these conditions were met, we positioned the predator model on the group's anticipated travel direction outside their visual range. We then continuously recorded their vocal behaviour and scored the presence of any non-vocal anti-predator behaviour on an all-occurrence basis (branch dropping, defaecation, vigilance). Duet and predator songs were recorded using directional microphones (Sennheiser MKH 815T & Sennheiser ME66) with windshields connected to a digital stereo recorder (Marantz PMD660; settings 44.1 kHz, 16 bits) from December 2012 to August 2014.

## Experimental protocol

Each group was tested once with each stimulus type, which resulted in a total of 12 trials (*N* = 6 duet songs; *N* = 6 predator songs, Table S1; minimum interval between trials:

1 week), all broadcasted before 12:00 local time (to match timing of natural duet song production). Prior to playback experiments, we measured the peak intensity of female great call climaxes in spontaneous duet songs (i.e. loudest notes, *Terleph, Malaivijitnond & Reichard, 2016*) at an estimated recording distance of 10–20 m using a REED ST-805 (REEDinstruments, Wilmington, NC, USA) sound pressure metre (frequency range 31.5 Hz–8 kHz, measuring level range 30–130 dB, 0.1 dB resolution, accuracy ± 1.5 dB). We measured three great call climaxes per female from the six song-providing groups (Table 1), which resulted in a mean sound pressure level of 78.2 ± 8.0 dB ($n = 18$; dB SPL, A-weighting sound pressure levels for general sound level measurements, and 125 ms fast time weighting). We then broadcasted songs such that subjects always heard recordings from one of their direct neighbours (Table S2), with comparable natural audibility (tested at each playback location with a decibel metre, matching climaxes SPL measurements, with real time adjustments in coordination with both experimenters depending on weather conditions on the testing day) and from spatially realistic locations 15–20 m within the canopy from where the song providing group had been seen before within the respective territories.

We standardised the distance between the speaker and subjects to about 150 m (mean ± SD: 149 ± 17 m), with playback conditions randomly counterbalanced (Table S2). Stimuli were broadcasted when the same conditions as for predator model presentation had been met, using a Climate CL60-T2 speaker connected to a Kenwood KAC-5203 amplifier, in conjunction with a Roland R-05 digital player.

Playback trials were carried out from spatially realistic locations, that is, from the home range of the song-providing group towards the home range of the target group. In doing so, we took a number of precautions such that the song-providing group could not overhear its own song. Before each trial, we ensured that the song-providing group was not in the vicinity of the speaker (>100 m radius). We then monitored the area for a period of 1 h to further ensure that the song-providing group was not nearby. For each trial, the speaker was positioned in the overlapping zone between the song-providing and target group, such that it was facing away from the home range centre of the song-providing group towards the target group.

### Data collection

Due to the difficult visual conditions in the forest, it was impossible to continuously video-tape the entire duration of trials nor to film all group members simultaneously. We therefore decided to restrict observations to the primary male of each group. Males are easily identifiable by their body hair colouration, facial features and genitals. Primary males were video recorded as long as possible (i.e. until they moved out of sight) using a Panasonic SDR-S26 Camcorder. Videos were coded using ELAN software (ELAN (V5.2) Nijmegen: Max Planck Institute for Psycholinguistics). Because the speaker location was not visible on the video clips (outside camera range) it was necessary for the experimenter to comment on the male's gazing direction during filming, which made blind coding redundant. All video recordings are available on figshare (https://doi.org/10.6084/m9.figshare.12363050.v1).

**Table 2 Behavioural response variables extracted for the primary males in both playback conditions.**

| | Definition |
|---|---|
| **Behaviour** | |
| Feeding | Handling or consuming food items |
| Resting | Prolonged stationary position, with or without eyes closed |
| Grooming | Auto- or allo-grooming (giver and receiver identity were collected) |
| Social | Mating, play, aggressive, or parental behaviour |
| Moving | Travel within or between trees (at least 2 metres) |
| Vigilance | Scanning the environment, head rotating by at least 45° (*Koenig, 1998*) |
| Other | Behaviour not classified into any of the above categories |
| **Body position** | |
| Hanging | Suspended in the air, grabbing a branch or a tree part with at least one arm |
| Sitting/Lying down | Sitting on a branch or on the ground / Resting in horizontal position |
| Gaze direction (staring at a specific location/direction/animal/person for ≥ 3s) | |
| Speaker | Staring in the direction of the speaker |
| Ground | Looking towards or actively scanning the ground |
| Canopy | Looking around, or towards a specific location in the trees at the same elevation as the animal location |
| Sky | Looking up at the sky |
| Group member | Looking at a group member (the identity of the receiver was collected) |
| Observer | Looking at the observer |
| Elsewhere | Looking in a direction that cannot be classified into any of the above categories |
| Nowhere | Resting with eyes closed |
| **Other measurements** | |
| Elevation (m) | Height of the animal in relation to the ground |
| Proximity (m) | Distance between the two focal individuals (paired male and female) |
| Defaecation/Urination | Exuding faeces and/or urine |
| Dropping branch | Individuals shaking branch(es) so as it ended up falling on the ground |
| Latency of first look towards the speaker (s) | Time elapsed between stimulus onset and first look towards the speaker |
| Duration of first look towards the speaker (s) | Duration of first gaze directed towards the speaker location |

Regarding long-term effects, we collected 5-min scan samples of the primary male's behavioural activities, gaze directions, body positions, elevations (m) and proximities to their female partner (m) during 1 h after each trial (i.e. 13 scans per trial; Table 2). Furthermore, we scored all defaecation/urination and branch dropping events over a two-hour period using all occurrence sampling.

## Vocal responses

We digitised, analysed and compared songs given in response to both playback conditions, using Raven Pro 64 1.4 (Cornell laboratory of Ornithology, Ithaca, NY, USA). For the introductory sequence, we determined the duration of the initial 'hoo' notes series (s) and the corresponding number of 'hoo' notes, the type of the first ten notes following the 'hoo' series, and the duration of the introductory sequence (i.e. latency to the first female

great call). We measured the interval between the female great call and the male coda reply (s), the total song duration (s), and determined whether a neighbouring group also produced a song and its type. Finally, we identified the presence of 'sharp wow' notes and we measured the latency to the first 'sharp wow' note (i.e. time elapsed in seconds between the onset of the song bout and the first 'sharp wow' emitted).

This study was approved by the School of Psychology Ethics Committee of St. Andrews University. Approval was given on the understanding that the ASAB guidelines for the Treatment of Animals in Behavioural research and Teaching are adhered to (n°16112011). The research permit was delivered by the National Research Council of Thailand (NRCT, n°0002/5841).

## DATA ANALYSIS

### Behavioural responses

We compared behavioural responses within subjects and across playback conditions; the primary male's latency and duration of first looks towards the speaker, the occurrence of defecations/urinations and branch droppings, the average distance to their female mate and the canopy heights (medians across all scan samples; Table 2). For categorical data (i.e. activity, body position and gaze), we summed up and calculated for each individual the proportion of each behaviour within the categories (see Table 2) and compared the behavioural pattern across playback conditions.

### Vocal responses

We compared the number of introductory 'hoo' notes and the duration of the introductory 'hoo' notes series, the number of other relevant 'hoo' and 'leaning wa' notes within the first ten notes following the introductory 'hoo' series, and the introductory sequence duration, within groups and across conditions. For the great call sequence, we compared male response delays to the female great calls. Finally, we compared the total song duration between playback conditions, identified the presence of 'sharp wow' notes, and measured the latency to first 'sharp wow' note produced.

### Statistical procedures

Due to small sample sizes we opted for non-parametric statistics. Wilcoxon matched-pair signed-rank tests were performed for behavioural data analysis, with exact significance levels reported (*Siegel & Castellan, 1988*; *Mundry & Fischer, 1998*). For vocal data, we used Kruskal–Wallis rank sum tests with a Benjamini & Hochberg procedure to correct for multiple testing (*Benjamini & Hochberg, 1995*). Post-hoc tests were either Wilcoxon rank sum tests with Benjamini & Hochberg $p$-value adjustments or *Dunn (1964)*'s tests with Benjamini & Hochberg $p$-value adjustments for eventual ties. To compare the type of the first 10 notes produced across contexts we used a Pearson's Chi-squared test followed by Chi-squared post-hoc tests with Benjamini & Hochberg p-value adjustments. Statistical analyses were performed using R V3.5.1 (*R core Team, 2018*) with the significance level set at 0.05.

## RESULTS

### Vocal behaviour

#### Response rates

In the duet song condition, 5 of 6 groups responded with duet counter-singing to playbacks of duet songs (Table S3). In addition, eight neighbouring groups that shared their borders with the song-providing group or the tested group also produced duet songs during 3 of 6 trials ($N = 3$, $N = 1$, $N = 4$ neighbouring groups, respectively, see Table S4), while none of them produced a predator song.

In the predator song condition, 1 of 6 groups responded with a predator song to playbacks of a predator song (within the first 10 min, see Table S3). The response song contained a highly delayed first great call and many 'sharp wow' notes, highly typical for a predator song. In addition, two distant (non-neighbouring) groups also produced predator songs during 2 of 6 trials, again characterised by a delayed first great call and 'sharp wow' notes (Table S4). None of the groups ever produced a duet song.

#### Song structure

Playbacks of duet songs reliably triggered synchronised singing by the mated pair of the target groups. To confirm that these vocal responses ($N = 5$) qualified as regular duet songs, we compared them to both spontaneously produced duet songs and experimentally induced predator songs (using a clouded leopard model; Table 3) by the same groups.

First, there were significant differences across all six variables tested (Table 3), while subsequent pairwise comparisons revealed significant differences between predator songs and the two other song types, but not between spontaneous duet songs and response songs elicited by playbacks (Table S5 for detailed pairwise comparisons).

Second, male latencies to reply to their female's great calls also differed significantly between song types ($\chi^2(2) = 33.90$, $P < 0.001$, $N = 82$, Kruskal–Wallis rank sum test). Here as well, post-hoc analyses revealed that males gave earlier replies to female great calls in the predatory context (mean delay: $-1.7 \pm 1.6$ s, $n = 20$) than in spontaneous duets ($0.6 \pm 0.7$ s, $n = 28$) or playback duet responses ($0.5 \pm 0.6$ s, $n = 34$) ($P < 0.001$ in both cases), with no difference between spontaneous and playback duet responses ($P = 0.530$; Dunn's post-hoc test for multiple comparisons, with Benjamini & Hochberg correction).

Finally, we compared the first 10 notes produced by males and females immediately following the introductory 'hoo' note series (mean duration: $10.50 \pm 2.8$ s, $n = 30$, accounting for a total of 100 notes per song type). Significant differences were found between song types regarding their early note composition in 'hoo' and 'leaning wa', but also in 'wa-oo' notes ($\chi^2(4) = 96.86$, $P < 0.001$, Pearson's Chi-squared test). Predator songs contained more 'hoo' notes and fewer 'leaning wa' notes than duet songs, with no differences between spontaneous and playback duet responses. However, 'wa-oo' notes were more common in playback duet responses than spontaneous duet songs, and again in spontaneous duet songs than predator songs (Table S6 for detailed pairwise comparisons).

**Table 3** Comparison of spontaneous duet songs ($N = 5$), predator songs ($N = 5$), and songs given in response to playback of duet songs ($N = 5$) by the same five groups (Kruskal–Wallis rank sum test).

| Variables** | Spontaneous duet song | Predator song | Response song | df | $\chi^2$ | P value* |
|---|---|---|---|---|---|---|
| Duration introductory 'hoo' series (s) | 8.0 ± 3.1 | 23.4 ± 6.7 | 4.7 ± 2.7 | 2 | 10.5 | <0.05 |
| N introductory 'hoo' notes | 11.0 ± 4.5 | 48.8 ± 14.4 | 7.4 ± 2.7 | 2 | 10.2 | <0.05 |
| Song duration (s) | 789.4 ± 294.8 | 2,396.4 ± 775.8 | 1,006.8 ± 122.3 | 2 | 10.2 | <0.05 |
| Latency to 1st great call (s) | 101.3 ± 33.5 | 816.4 ± 368.0 | 99.0 ± 41.1 | 2 | 9.5 | <0.05 |
| Latency to 1st 'sharp wow' (s) # | 78.1 ± 31.1 | 370.5 ± 183.2 | 90.8 ± 35.9 | 2 | 9.0 | <0.05 |
| N 'sharp wows' | 9.2 ± 8.0 | 362.2 ± 233.9 | 5.6 ± 6.0 | 2 | 9.8 | <0.05 |

Notes:
# Kruskal–Wallis rank sum test for $N = 14$ songs (W did not produce any 'sharp wow' notes in spontaneous duet; $N_{duet} = 4$, $N_{predator} = 5$, $N_{response} = 5$).
\* $P < 0.05$ corrected.
\*\* Means ± SD.

## Non-vocal behaviour

We were able to record the immediate behavioural responses of primary males in 5 of 6 groups (Table 1). All males responded by turning their heads towards the speaker, albeit with no latency differences across playback conditions (median duet: 1.1 ± 1.8 s, predator 2.8 ± 3.3 s, $V = 2$, $P = 0.188$, $N_{duet} = 5$, $N_{Predator} = 5$, Wilcoxon matched-pair signed-rank test, Fig. 4A). Additionally, we found a trend (although not significant) towards longer gaze duration in the predator than the duet song condition (median duet: 2.0 ± 1.4 s, predator: 12.6 ± 6.1 s, $V = 0$, $P = 0.063$, $N_{duet} = 5$, $N_{Predator} = 5$, Wilcoxon matched-pair signed-rank test, Fig. 4B).

For long-term behavioural responses, we collected data on all six primary males and found no differences across playback conditions in grooming, resting and displacement activities but a significant difference in feeding, with individuals less likely to engage in feeding activities after predator than duet song playbacks (Table 4). Regarding anti-predator behaviours, we found no differences in canopy use, distance between mates, and number of branch droppings across conditions. However, males were more vigilant and defaecated significantly more often following predator compared with duet song playbacks (Table 4).

Following playback of a predator song, males increased their vigilance activity (Fig. 5), directed more gazes towards the ground (Fig. 6A) and less towards the upper canopy (Fig. 6B) compared with duet treatment (Table 4).

# DISCUSSION

## Summary

White-handed gibbons produce two structurally distinct songs in context-specific ways; duet songs (in non-predatory contexts) and predator songs (to clouded leopards and other predators). The two song types differ in the overall duration, frequency and distribution of specific notes ('hoo', 'leaning wa', 'sharp wow') and in the location of

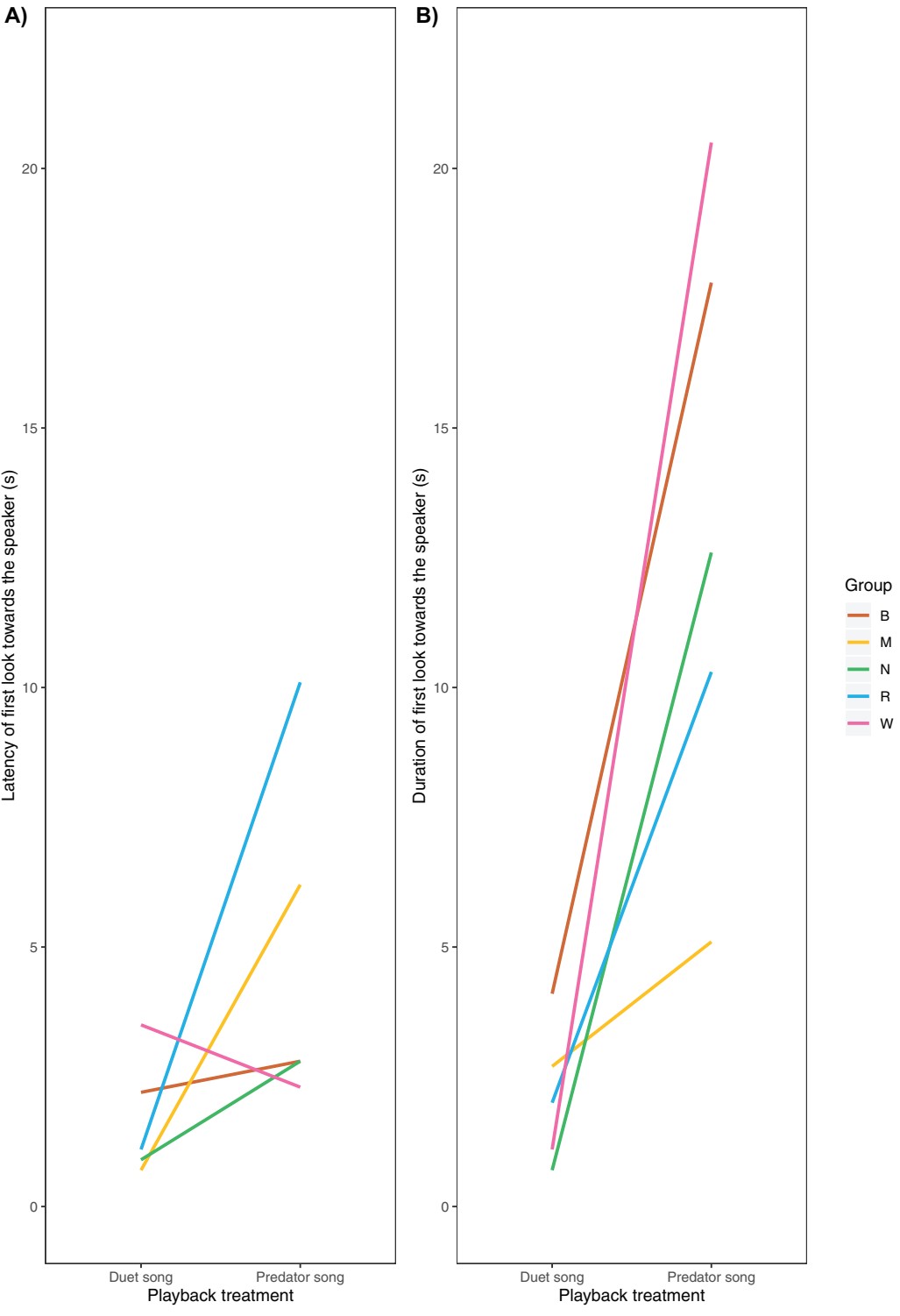

**Figure 4** (A) Latency and (B) duration of the male gibbons' first gaze towards the speaker broadcasting a simulated neighbouring group's song (duet vs. predator song condition).


**Table 4 Comparison of male long-term behavioural responses between playback treatments (Wilcoxon matched-pair signed rank tests, N = 12 playback trials, with a total of n = 156 scan sampling observations, i.e. 13 scans per individual for 1 h).**

| Variables** | | Duet song playback | Predator song playback | V | P value |
|---|---|---|---|---|---|
| Behavioural activity | Grooming | 1.0 ± 1.6 | 1.7 ± 2.0 | 5.5 | 0.688 |
| | Moving | 2.0 ± 1.3 | 1.7 ± 1.5 | 13.5 | 0.594 |
| | Resting | 0.8 ± 0.8 | 0 ± 0 | 10 | 0.125 |
| | Feeding | 4.0 ± 1.4 | 0.3 ± 0.8 | 21 | <0.05* |
| | Vigilance | 2.2 ± 1.3 | 9.0 ± 2.0 | 0 | <0.05* |
| Body position | Hanging | 7.2 ± 2.5 | 5.0 ± 1.7 | 3.5 | 0.188 |
| | Sitting/lying | 5.8 ± 2.5 | 8.0 ± 1.7 | 17.5 | 0.188 |
| Gaze direction | Speaker | 3.0 ± 1.7 | 4.5 ± 1.1 | 2 | 0.125 |
| | Canopy | 8.7 ± 1.2 | 3.2 ± 2.5 | 21 | <0.05* |
| | Ground | 0 ± 0 | 5.0 ± 1.3 | 0 | <0.05* |
| | Group member | 1.3 ± 1.5 | 0.3 ± 0.8 | 8.5 | 0.375 |
| Elevation (m) | | 17.6 ± 6.2 | 25.1 ± 7.1 | 3 | 0.156 |
| Proximity to mate (m) | | 8.9 ± 7.3 | 10.3 ± 7.7 | 7 | 0.563 |
| Dropping branch[†] | | 0 ± 0 | 0.5 ± 0.8 | 0 | 0.5 |
| Defaecation/Urination[†] | | 0.3 ± 0.5 | 3.2 ± 1.2 | 0 | <0.05* |

Notes:
[†] All occurrence behaviours recorded over 2 h post trial.
* $P < 0.05$.
** Means ± SD.

the female great calls and male replies within each song. In this study, we investigated whether individuals discriminated between these two structurally different song types and whether they could infer meaning from them. We found several lines of evidence in favour of such an ability. First, playbacks of duet songs reliably elicited natural duet song replies (identifiable by several acoustic parameters) in neighbouring groups and in more distant groups, similar to how natural duet song spread throughout the forest (*Raemaekers & Raemaekers, 1985b*; J. Andrieu, 2012–2014, unpublished data a). Second, playbacks of predator songs never triggered duet songs in any group, but occasionally predator song replies (identifiable by several acoustic parameters) in one of six neighbouring groups and two non-neighbouring distant groups. Finally, subjects consistently showed anti-predator behaviours (vigilance, ground scanning, defaecation) and a tendency for longer first look towards the speaker after predator compared to duet song playbacks. Based on these data, we concluded that white-handed gibbon song conveys key information about the world, which is made accessible to recipients by a number of structural regularities. This conclusion fits with previous research by *Clarke, Reichard & Zuberbühler (2006)* who first demonstrated the presence of structural differences in white-handed gibbon songs.

## Singing as anti-predator behaviour

Similar to other large cats, clouded leopards are opportunistic predators that attack both terrestrial and arboreal species, including primates (*Rabinowitz, Andau & Chai, 1987*;

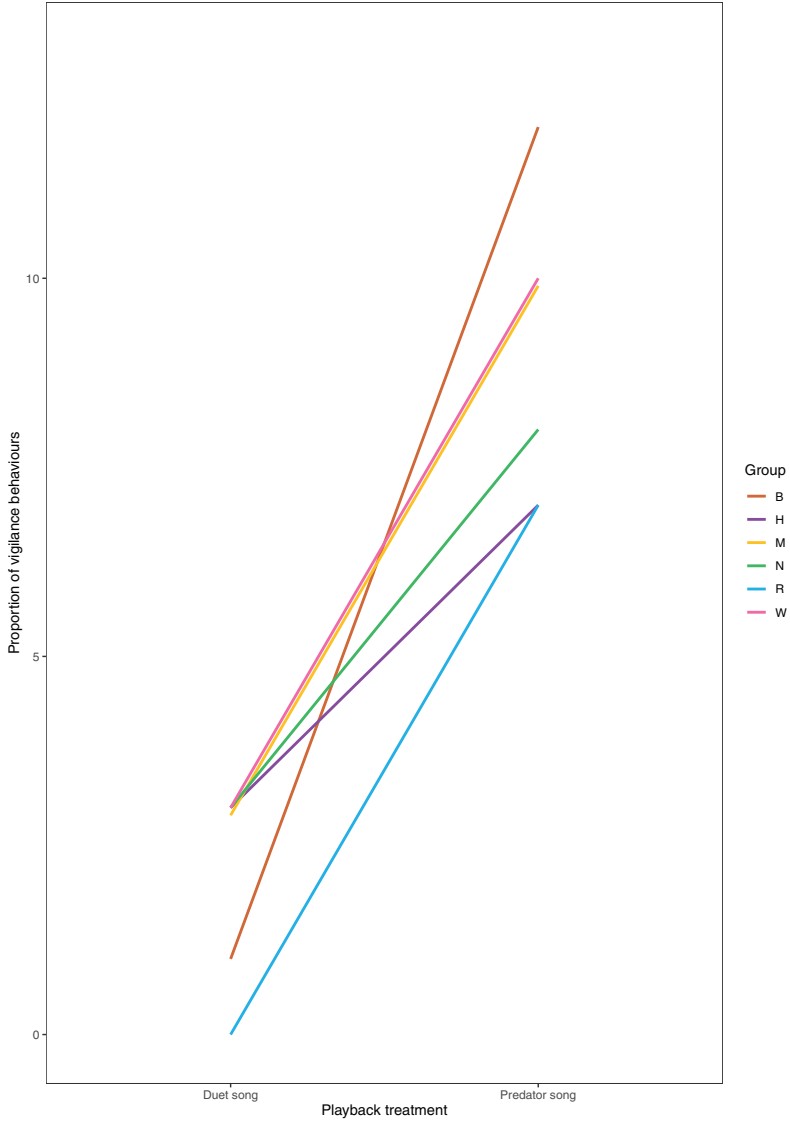

**Figure 5 Proportion of vigilance behaviours displayed by males in each playback condition (*N* = 6 males).**

*Grassman, 2001*). Hence, a somewhat surprising finding was that subjects remained mostly silent to others' predator songs, despite showing strong anti-predator behaviour (males and females appeared to behave in the same way, i.e. ground scanning, vigilance, defaecation). The lack of vocal response may be part of a cryptic strategy to conceal the group's location when a dangerous stalking predator is presumed in the vicinity (*Aguilar de Soto et al., 2012*; *Grow, 2019*). However, this does not explain why 1 of 6 target groups and two distant groups still responded with predator songs to the playbacks. It is possible that gibbons pursue a flexible vocal strategy, altering between 'crypsis' and 'perception advertisement' depending on perceived personal risk, the ability to benefit neighbouring relatives, and the likely dissuasive effect on the predator itself (*Zuberbühler, Jenny & Bshary, 1999*; *Clarke, Reichard & Zuberbühler, 2006*).

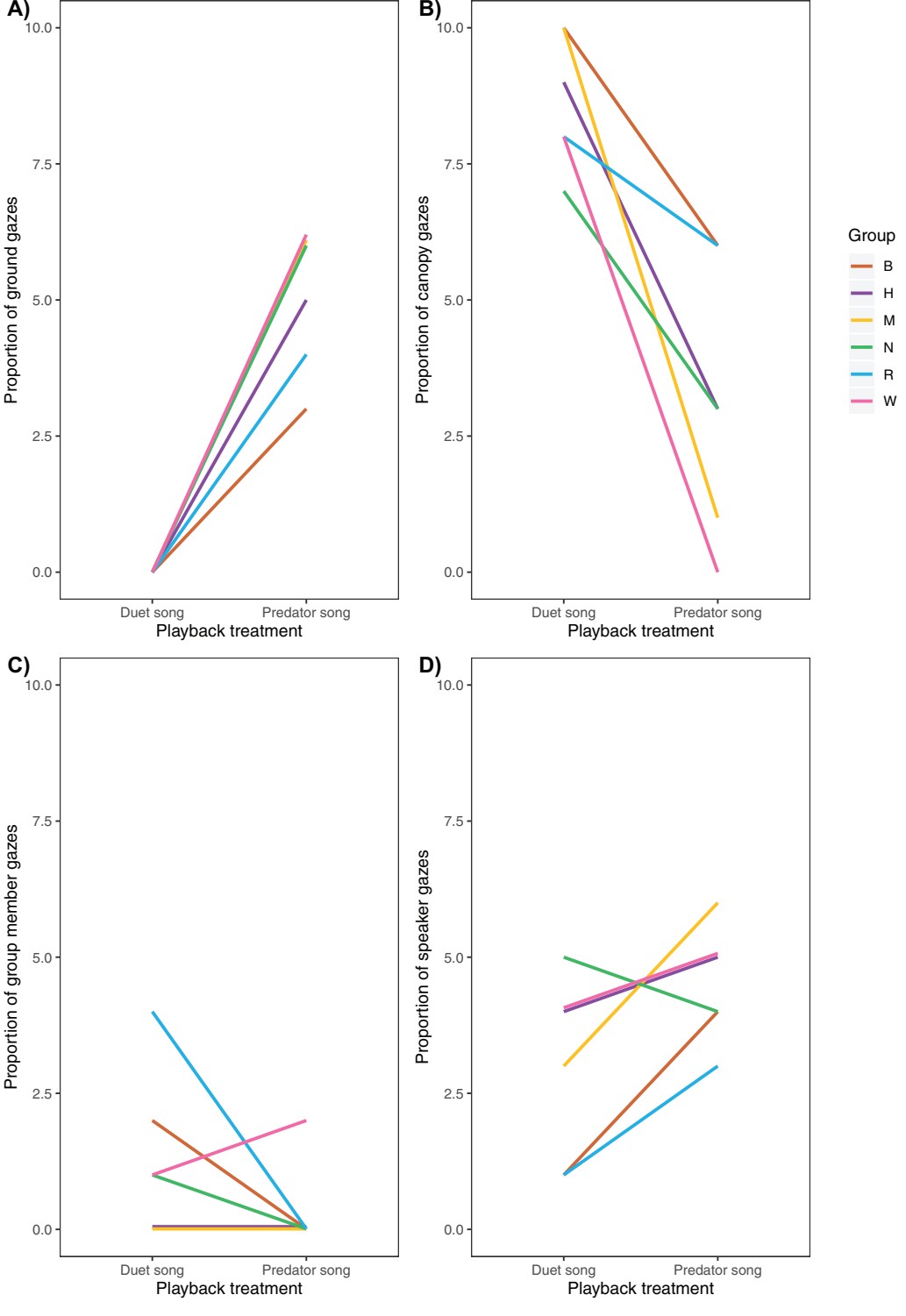

**Figure 6 Variation of (A) ground, (B) canopy, (C) speaker and (D) group member gazes between playback treatments (N = 6 males).**

Equally relevant is the fact that the three predator song responses were shorter than natural predator songs (Tables S3 and S4). We can think of several explanations for this finding. First, as mentioned already, it is possible that groups tried to minimise their own exposure to the predator if they decided to respond to another group's predator song. Second, differences in predator song duration may function as indicators for perceived urgency, with longer songs indicating more serious threats than shorter songs. We find this less likely to be an evolved function since listeners would have to wait for (and compare) considerable amounts of time periods before extracting the relevant information. Finally, differences in song duration may be linked to how callers perceive the predator (visually, linked to mobbing the predator vs. acoustically, linked to localising the predator). A Direct observation of a real encounter with a tiger is in line with this hypothesis (*Uhde & Sommer, 2002*). In this instance, group A uncommonly travelled backward towards the tiger's location (spotted 50 m away) and sang for at least 1 h and a half, suggesting that singing primarily serves first and foremost as a predator deterrence device and second as a conspecific warning signal if the exact location of the predator is unknown and groups feel reasonably safe.

## Singing as territorial behaviour

In related research (J. Andrieu, 2012–2014, unpublished data), we have shown that spatial proximity between two neighbouring groups tends to lead to duet song overlap, due to the fact that the second group refuses to delay singing until the first group has finished their duet song. This behaviour is attenuated by kinship, to the effect that related individuals are more likely to respect each other's duets, even if produced at close distances. In the current study, all study groups started producing duet songs while the playback duet song was still being broadcast, suggesting that the manipulation was perceived as a territorial threat. Unfortunately, we could not statistically analyse the effect of genetic relatedness in this study because the sample size was too small ($N = 6$ groups).

## Singing as compositional behaviour

Although our study has focussed on song comprehension, it has also generated a more detailed picture of the structural composition of white-handed gibbon songs. *Clarke, Reichard & Zuberbühler (2006)* already noted that the duet songs of gibbon groups that were not well habituated to human observers contained elements that were normally found in predator songs, notably 'sharp wows'. In our study, all groups were fully habituated to human presence, yet some groups still produced 'sharp wow' notes in their duet song replies to playbacks of neighbouring duet songs, but also in 4 of 5 natural duet songs (Table 3), of which 3 were involved in duet counter-signing exchanges with previous duetting direct neighbours. Another structural subtlety concerned the use of 'wa-oo' notes. This note type was near absent in predator songs but common in the early parts of the duet songs, especially the ones given in response to duet song playbacks. We attribute these findings to the fact that our experimental design consisted of playbacks of song recordings at relatively close distances (about 150 m), which may have been

perceived as a social threat by some groups, either territorial or risk of partner defection. Future work is required to test whether these notes are actively used to describe events in hierarchically structured ways (main: predatory threat y/n; subsidiary: social threat y/n), similar to how humans represent natural events as tree structures in both cognition and language (*Zuberbühler, 2019b*).

## CONCLUSION

Gibbons play an interesting role in questions about the biological roots of language-related capacities in humans. Although part of the Hominoidae family, they maintain a relatively basal position in their phylogeny by diverging from the great apes some 16 million years ago (*Carbone et al., 2014*). Nevertheless, gibbons show interesting vocal behaviour by which a small repertoire of acoustically distinct notes are combined into higher-order structures, such as figures, phrases and sequences, assembled into different song types (*Raemaekers, Raemaekers & Haimoff, 1984*; *Clarke, Reichard & Zuberbühler, 2006*). These findings have some implications for the ongoing debate about syntax and phonology in animal communication (*Bolhuis et al., 2018*; *Townsend et al., 2018*).

In a previous study (*Clarke, Reichard & Zuberbühler, 2006*), structural differences between gibbon song types were explained as a case of animal syntax although this was based on a very broad definition of the term. An alternative, more restricted definition of syntax invokes semantics, notably that the units subjected to syntactic operations (e.g. the notes) are meaningful, for which there is currently no evidence in gibbon song.

Whatever definition is applied, gibbon song has several levels of complexity and future research should be directed at the acoustic variation in the different note types and their combinations. For example, in the current study we found that the production of 'wa-oo' and 'sharp wow' notes might be linked with perceived social threat. So far, systematic analyses have been restricted to the early parts of the song (based on the assumption that predator information should be conveyed early on) with individual contributions not systematically studied. Traditional acoustic analysis may not suffice to make meaningful progress, suggesting that automated call extraction and categorisation techniques may offer more promise to explore the full combinatorial, hierarchical and compositional capacity of gibbon song (*Kershenbaum, 2014*; *Kershenbaum et al., 2014*, *2016*; *Kershenbaum & Garland, 2015*; *Fedurek, Zuberbühler & Dahl, 2016*).

## ACKNOWLEDGEMENTS

We thank Melanie Jackson, Prayoon Saenkhot, Attaklab Chaiyawat, and Surasak Homros for their help in the field.

### Funding

This research project has been funded by the Leverhulme Trust (Research Leadership Award F/00268/AP), the European Research Council (grant number FP7; PRILANG

GA283871) and the Swiss National Science Foundation (310030_185324). The funders had no role in study design, data collection and analysis, decision to publish, or preparation of the manuscript.

## Grant Disclosures

The following grant information was disclosed by the authors:
Leverhulme Trust: F/00268/AP.
European Research Council: FP7; PRILANG GA283871.
Swiss National Science Foundation: 310030_185324.

## Competing Interests

The authors declare that they have no competing interests.

## Author Contributions

- Julie Andrieu conceived and designed the experiments, performed the experiments, analysed the data, prepared figures and/or tables, authored or reviewed drafts of the paper, and approved the final draft.
- Samuel G. Penny performed the experiments, authored or reviewed drafts of the paper, and approved the final draft.
- Hélène Bouchet conceived and designed the experiments, authored or reviewed drafts of the paper, and approved the final draft.
- Suchinda Malaivijitnond conceived and designed the experiments, authored or reviewed drafts of the paper, administered and coordinated official and institutional support, and approved the final draft.
- Ulrich H. Reichard conceived and designed the experiments, authored or reviewed drafts of the paper, supplied long-term demographic records and administered and coordinated official and institutional support, and approved the final draft.
- Klaus Zuberbühler conceived and designed the experiments, authored or reviewed drafts of the paper, and approved the final draft.

## Animal Ethics

The following information was supplied relating to ethical approvals (i.e., approving body and any reference numbers):

School of Psychology Ethics Committee of St. Andrews University provided approval for this research. Approval was given on the understanding that the ASAB guidelines for the Treatment of Animals in Behavioural research and Teaching are adhered to (16112011).

## Field Study Permissions

The following information was supplied relating to field study approvals (i.e., approving body and any reference numbers):

The research permit was delivered by the National Research Council of Thailand (n°0002/5841).

## Data Availability

Raw measurements are available in the Supplemental Files. The large files (videos and song examples) are available at figshare: Andrieu, Julie; Penny, Samuel G.; Bouchet, Hélène; Malaivijitnond, Suchinda; Reichard, Ulrich H.; Zuberbühler, Klaus (2020): White-handed gibbons discriminate context-specific songs compositions. figshare. Media. DOI 10.6084/m9.figshare.12363050.v1.

## Supplemental Information

Supplemental information for this article can be found online at http://dx.doi.org/10.7717/peerj.9477#supplemental-information.

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
