# Peer review of "White-handed gibbons discriminate context-specific song compositions"

_PeerJ, doi:10.7717/peerj.9477_

## Round 0.1 · original submission · Minor Revisions

Thank you very much for your submission to PeerJ. I was fortunate to receive reviews of your article from three experts in the field. The reviewers commented on the clarity of your writing and the timely nature of your study; I concur and I believe that your article will likely be suitable for publication in PeerJ following some minor revisions.

All three reviewers provided thoughtful and detailed feedback and I encourage you to respond to each of their comments as you prepare your revisions. I will not reiterate the reviewers' comments here but I did want to highlight that a common point of concern regarded the clarity of your methods and I encourage you to add the detail requested by the reviewers. Furthermore, the reviewers asked that you expand on and clarify some of your introductory and discussive text and I also agree that that would be worthwhile. Lastly, given the electronic nature of this journal, without page restrictions, I agree with the reviewers that some of your supplemental materials could be included in the main body of your article. In particular, I think your article would benefit from the inclusion of the current Table S1 and Figure S2 in the main body of the text.

I look forward to receiving your revision.

Reviewer 1 ·

Basic reporting

The ms is concise and well written, with good coverage of the literature, and is structured appropriately. Part of the rational for the study, used in interpretation, involves two mss "in press", so are not available. In that sense, the ms is not fully self-contained.

A few minor comments on writing:

Line 57. Explain what these studies show.
63. "In linguistics"... OK, but what do your mean by "pragmatics" here.
160. what is the measure of variation?
202. Duration of notes, or duration of a series of notes?
258-262. Cut; this just repeats the Introduction.
268. "waoo" does not appear in the main text (or S4), as I recall, until this point. Yes, it is in the Supplement, but you need also to include it in the main text.
272. Join this "paragraph" to the previous one, which has a topic sentence also including this material. Possible trim the resulting longer para.

I'd really like to be able to hear some recording of the songs (they are a highlight of SE Asian forests), and strongly recommend a couple of examples of duet and predators songs in the Supplement.

Experimental design

The design is generally clear and appropriate, with good equipment and rigorous definitions of responses. The sample size is small, which limits some conclusions, but this is not unusual in studies of primates. Furthermore, despite the small sample, there are clear differences in the response to playback treatments.

I noticed three things missing in the methods: (1) There is no reporting of playback amplitude. The text 156-157 states that the playbacks were broadcast such that the "subjects always heard the recordings", and "with comparable natural audibility (tested at each playback with a decibel meter)." Without presenting data on the actual playback amplitudes, amplitude of real calls, or information on gibbon hearing/responses, none of these claims is currently supported. Please give appropriate data, or explain why it is not given and why you think it is OK not to give the data. (2) Responses to playback were videoed, which often gives the opportunity for blind scoring, but it is not explicitly stated that the videos were not scored blind (which I assume was the case). Please say clearly if scoring was blind or not, and if not explain why is was not done blind. (3) I assume that focal animals were sometimes out of sight during the videos. What did you do in that situation?

Validity of the findings

The rationale for the study is clear. The general finding, that there were behavioural differences in response to the two treatments appears sound (although I would like to know the answer to both methodological omissions above). The data are provided. The conclusions are stated and for the most part seem robust.

The supplement Table S4, and associated statistical analyses and conclusions are not fully supported. Failing to reject a null hypothesis of no difference, especially with small sample sizes, does not show that the playback duet responses and spontaneous duets are the same (and indeed one difference was found). Please modify the claims and conclusions about these analyses. S4 & analyses would be much more compelling if it also included quantitative data on predator songs, and so revel that responses to playback more closely resemble natural duets and than predator songs. I imagine that there would be some overlap with the paper "in press", but I also expect that the overlap would be minimal and so may not compromise those papers. I think Tables S4 would be better in the main results.

283- on. Gibbons looked down after playback of the "alarm" duets given to leopard prompt. This seems to imply that they have different alarms to terrestrial and aerial predators. Is this the case? And if they always look down, as if expecting a terrestrial predator, why bother being cryptic? A leopard couldn't chase a gibbon in the trees. Are gibbons also hunted by people? That might explain crypsis, much like in the Diana monkey findings.

305-307. Yes, but the lack of evidence might be because people have not looked at acoustic structure of notes. Without an analysis of the acoustic structure of notes, &/or appropriate experiments, you'll surely never be able to be sure that "syntax" is responsible for communicating meaning. That point could perhaps could be addressed more directly.

313-314. The claim the duet and predator songs differ is NOT shown in Table S3. See point about adding quantitative data on predator songs to Table S3.

319-325. It might be difficult to compare notes, but it seems worth trying, for reasons already mentioned.

331. This ms does not show the "complex coding abilities". I guess that will be in the "in press" articles?

Additional comments

I enjoyed reading the paper; it's nicely written and interesting.

Reviewer 2 ·

Basic reporting

all fine, although some suggestions for additional background information: please see 'General comments' below

Experimental design

please see 'General comments' below

Validity of the findings

please see 'General comments' below

Additional comments

The manuscript entitled: ‘White-handed gibbons discriminate context-specific songs composition‘ by Andrieu et al., tests the hypothesis that gibbons respond differently to the playback of duet songs versus predator songs, suggesting that gibbons perceive a difference in these song types. The authors suggest that this could be a result of the difference in note prevalence or timing of the songs, given that the two songs share syllable types. Playbacks in the field are difficult but particularly relevant to natural behavior. The authors further took care to match playbacks to the natural sound level and also the general location of natural calls in the canopy. They measured a range of vocal and nonvocal responses to playback, creating a richer profile of call-responsive behavior. The results are consistent with white-handed gibbons distinguishing different types of songs from neighbouring groups. The manuscript could be made more accessible to non-specialist reviewers by fleshing out the information included in the manuscript.

Comments:

Although presenting playbacks in the field is a challenging undertaking, why did the authors choose to present playbacks to 6 rather than all 8 of the groups they work with? Please describe this in the Methods.

Some of the information in the early supplemental figures (esp 1 and 2) would be useful in providing context in the body of the manuscript- can these be moved? Also, it would be helpful if supplemental Fig 1 included labels of all of the types of notes (the ‘waoo’ would be especially interesting).

In fig 1, it would be helpful to provide some kind of indication of time- do the songs take tens of seconds? Minutes? Additionally, it is difficult to find any information about the variability in the song sequences used for playback. There is some information in supplemental table 4 on the duet call, but I did not see information of the variability among predatory song playbacks. This information would be helpful in understanding the range of parameters that gibbons heard across playbacks.

Some of the information in the supplemental text would be interesting as part of the main manuscript, especially what is known about the function of duets from past work. Further, are there any functional studies of predator calls? Are the predator songs directed towards predators (as the wording here implies), for distraction, warning, what exactly? Are they a sort of alert to conspecifics for predator presence?

Why were only male nonvocal responses to playback measured, and not those of females? How was the primary male identified? Please state in the Methods

Methods: How much time passed between duet versus predatory playbacks for a given group? Were they on the same day? Separated in time?

What time of day they did playbacks occur? The introduction (line 82) states that “…mated pairs produce morning duets…” Is time of day significant for this specific vocal production?

Regarding the Bonferroni corrections, how many repeated comparisons were made and corrected for?

Given that the responses to duet playback occur before the duet songs are over, the authors make the point that neighbours must be responding to early parts of the duet song, potentially lending more weight to the difference in ‘ooaa’ syllables in allowing groups to distinguish duet from predatory songs. This is a very interesting point, and highly relevant to the manuscript’s premise that white-handed gibbons can distinguish among the two song types. Isn’t this information worth including in the main part of the manuscript?

Supplementary text, lines 47-50: ‘However, songs in response to duet playbacks contained more ‘waoo’ notes than spontaneous duets. Despite the early ‘waoo’ note composition difference, elicited duets in response to playback treatments could not be distinguished from natural duet songs.’ These two statements seem contradictory to each other- what am I missing?

Minor comments:

lines 104-105: the predictions at the end of the introduction are rather vague and could likely be made more specific (for example, what types of behavioural responses would be ‘adequate to the situation’ for predator or duet playbacks?).

line 138: what criteria were used to determine that a specific song was suitable for playback?

lines 222-223: This sentence is a little confusing: ‘In addition, eight non-neighbouring groups also produced duet songs in 3 of 6 trials, while none of them produced a predator song’ Does it mean that, of the eight groups, all eight produced duets in each of the 3 trials? Across these three trials? Other?

If non-neighboring groups responded to playbacks, were there any discernible differences in response based on proximity vs distance? For example, were closer groups more likely to respond?

·

Basic reporting

The basic reporting for this article is of very high quality. Overall, this is was an interesting paper that contributes new insights into the communication system of white-handed gibbons. The authors rightly point out that gibbons are a bit of a blank spot in our knowledge of Hominoidae. I appreciate that this paper looks at responses to the songs and the contexts in which they were produced – and it seems to tie in with another study that includes predator presentations.

My main suggestions are to include more details in the methods, and to elaborate on some of the theoretical points in the introduction and discussion. The paper is very concise, but at some points that conciseness means that the reader doesn’t get all of the required information. I've made specific comments to the author in areas that I think require more detail.

Experimental design

The experimental design is based on well-established methods within this field, and was appropriate for the question being asked. The sample size was quite snall, but that is to be expected in playback studies. There were a couple of points that needed more detail in order for others to replicate this study, which I've included in specific comments to the author.

Validity of the findings

The authors provide all of the data required for their analyses, and their results are clearly stated. There was some phrasing in the discussion that I felt would have required a different analysis (i.e. acoustic comparison of call elements for each song type) in order to substantiate those claims, and have made alternative suggestions in the comments to authors. The authors later acknowledged the current methodological limitations of gibbon song analysis, and so it would be good to tie those two points together.

Additional comments

Introduction
General - Do you have any idea about the rough frequency with which gibbons hear the songs of their neighbours? It’s not crucial to this study, but would be cool if there’s a reference so that we can understand the ecological implications/potential adaptive value of overhearing different song types.
Line 53: What does it mean that the variations in calls are “meaningful to recipients”? It would be useful to the reader if you could explain and give a specific example (in addition to the species list that you already have).
Line 64: It is my understanding that the Gricean approach to meaning is not that “the signaller’s intention affects signal meaning”, but rather that a signal has to be intentional in order to have meaning in the first place. I think the authors are trying to say that we need to consider the context in which calls occur, and are perhaps arguing for a more Carnapian approach? Plonking Grice at the end is just a bit confusing for the reader, unless you could add a sentence to wrap up the paragraph saying which approach you favour.
Line 68: This paragraph is great and a good model for the types of examples that would be helpful to the reader in the previous two paragraphs. Clear writing and interesting content!!
Line 85: “its production is subject to social influences”, such as … - please elaborate on which social factors impact song production in which ways. This information will be useful for interpreting your results later.
Line 90: I didn’t realise that gibbons have predator songs. That’s really cool!
Line 101: To talk about meaning in the Gricean sense, you need to establish that the songs are intentional. That’s why I wonder whether you should say you’re taking a Carnapian approach to pragmatics, then you can talk about meaning without hashing out the Gricean argument.

Methods
Line 111: Are gibbons in groups of 1 adult male, 1 adult female and their offspring? A brief description of typical group structure in the intro would give the reader some context. It could also tie into an explanation later (line 166) of why you chose to target the primary male for video recording. I see that you’ve given the specific group details in a supplementary table, which is also helpful!
Line 134: Figure 1 is a very helpful visualisation of the two song types.
Line 141: This is a solid set of criteria! It must have been challenging to collect these stimuli. Well done!
Line 155: Were you able to ensure that the individuals from which you took the song recordings were not able to overhear the recordings of their own voices? i.e. was someone aware of the location of the neighbouring group while you were conducting the playback with the target group? This has ethical implications, but also affects whether the target group may have been aware of the neighbours in a different direction from the one you presented the playback. If you were unable to know their location, it is important to state that here.
Line 166: Why did you target the primary male for video recording?
Line 166: Which video recorder did you use?
Line 167: Which program did you use to code the videos? What variables did you code? It’s too vague for someone else to replicate.
Line 169: For a moment, I thought that you were extracting this information from the videos. For clarity, could you start the sentence with “We also conducted a focal scan of the primary male during one hour following the playback (5-min scan sampling, i.e. 13 scans per experiment; Table 1),…”
Line 176: Is this in the video coding scheme, or an audio coding? It’s a bit unclear where this data is coming from.

Results:
Overall the results are presented in a very clear way that is easy for the reader to follow.
Line 243: Please include the direction of the difference in text, so that readers don’t need to interpret the table right away.

Discussion:
General – at some point in the discussion, there should be a note saying that all of these responses are from the primary males, and another study would be necessary to see whether female gibbons also respond appropriately to the different duets (you could also hypothesise that they will or will not, based on preliminary observations).
Line 272: The study demonstrates that gibbons respond differently to each type of duet, but it doesn’t provide direct evidence that the structure of the duets leads to these different responses. The authors describe that the calls are the same but in different lengths and different orders, but there is no direct comparison of the call types within the sequences that were used as stimuli. (I apologize if this was included in a reference to another study earlier and I’ve missed it – if so, could bring it up again here?)
Line 279: “discriminate and respond appropriately to the context in which another group’s song was produced” – I’m not sure the wording properly expresses what’s happening, it sounds like they are directly responding to the context. Perhaps something like “Our data show that gibbons are able to use another group’s song to determine the context in which that song occurred, as demonstrated by the listeners responding appropriately to the context of the callers. They discriminate the context of the call without having witnessed the context themselves.”
Line 284: “once in visual contact” with a predator?
Line 291: I think “sometimes” would be more appropriate than “temporarily”
Line 323: Oh, this seems to tie back to my previous point about direct comparison of elements within each duet type. Could you elaborate in a sentence after this one about why it will be important to use these analyses?

---

## Round 0.2 · Minor Revisions

Thank you very much for submitting your revised article to PeerJ. I have now read through your response to the reviewers’ feedback and also your revised article. Thank you for making such thorough edits. Accordingly, I do not feel the need to send it back out for review, however I do have a few points that I ask you to address before I can recommend your article for publication:

1. Please make the abstract a single paragraph
2. When first describing a species, please be consistent in providing the Latin species name (i.e. compare line 44 with line 55)
3. Please refer to the PeerJ guidelines on how to cite “in prep” work (e.g., Andrieu et al) as it must be cited as “unpublished” not “in prep”: https://peerj.com/about/author-instructions/
4. Line 108, for the first sentence of the final paragraph of your introduction, as you’re starting a new paragraph for clarity please state the song types specifically (i.e. duet and predator) rather than simply saying “song types” without clarification
5. Line 159 when describing how you recorded the duet calls here please provide details about the hardware that you used as you have done for the other call recordings.
6. Please make the opening two paragraphs of your discussion (ie lines 355-373) into a single paragraph

Also, I could access the fig.share link you included in your article but not the dryad link (https://doi.org/10.5061/dryad.vx0k6djnx) you provided in your response to the reviewers. Please double check this.

I look forward to receiving your revisions.

---

## Round 0.3 · accepted · Accept

Thank you very much for making these final outstanding revisions. It is my pleasure to recommend your article for publication in PeerJ.